# HybriDC: A Resource-Efficient CPU-FPGA Heterogeneous Acceleration System for Lossless Data Compression

**DOI:** 10.3390/mi13112029

**Published:** 2022-11-19

**Authors:** Puguang Liu, Ziling Wei, Chuan Yu, Shuhui Chen

**Affiliations:** College of Computer Science and Technology, National University of Defense Technology, Changsha 410073, China

**Keywords:** lossless data compression, CPU-FPGA heterogeneous acceleration, resource efficiency, performance modeling, LZ4

## Abstract

Lossless data compression is a crucial and computing-intensive application in data-centric scenarios. To reduce the CPU overhead, FPGA-based accelerators have been proposed to offload compression workloads. However, most existing schemes have the problem of an imbalanced resource utilization and a poor practicability. In this paper, we propose HybriDC, an adaptive resource-efficient CPU-FPGA heterogeneous acceleration system for lossless data compression. Leveraging complementary advantages of the heterogeneous architecture, HybriDC provides a universal end-to-end compression acceleration framework with application compatibility and performance scalability. To optimize the hardware compression kernel design, we build a performance–resource model of the compression algorithm taking into account the design goal, compression performance, available resources, etc. According to the deduced resource-balanced design principle, the compression algorithm parameters are fine-tuned, which reduces 32% of the block RAM usage of the LZ4 kernel. In the parallel compression kernel implementation, a memory-efficient parallel hash table with an extra checksum is proposed, which supports parallel processing and improves the compression ratio without extra memory. We develop an LZ4-based HybriDC system prototype and evaluate it in detail. Our LZ4 compression kernel achieves state-of-the-art memory efficiency, 2.5–4× better than existing designs with comparable compression ratios. The evaluation of total resource utilization and end-to-end throughput demonstrates the excellent scalability of HybriDC. In power efficiency, the four-kernel HybriDC prototype achieves a threefold advantage over the standard LZ4 algorithm.

## 1. Introduction

Nowadays, data are rapidly generated all the time. Large volumes of data bring significant overheads in data transmission, storage, and processing. As an effective way to reduce the data size, lossless data compression has been widely applied in data-centric scenarios [1,2,3]. However, for any lossless data compression algorithm, its throughput, compression ratio, and computing overhead must be traded off against one another. High-throughput algorithms will inevitably demand heavy computing overhead. There is a growing interest in offloading the computing-intensive workload to a field-programmable gate array (FPGA) [4,5,6,7]. Compared to CPU-based designs [3,8], FPGA accelerators can achieve higher throughput by better parallel pipeline processing. Thanks to reconfigurability, FPGA-based designs are more adaptable than the ASIC-based designs [9,10], which can save much time and investment cost. As a possible trend beyond Moore’s law, the CPU-FPGA heterogeneous architecture has become attractive in computing acceleration for its flexibility and potential performance [11,12,13].

Prior studies proposed many FPGA-based acceleration designs for lossless data compression [14,15,16,17]. These designs aimed to realize a high-throughput hardware compression kernel. However, the hardware resource usage and the design practicability were sometimes ignored. The main on-chip resources of FPGAs include lookup tables (LUTs), flip-flops (FFs), and block RAMs. These resources are distributed in fixed numbers and proportions on the FPGA chip. As shown in Table 1, most data compression acceleration designs have the problem of an imbalanced resource utilization. The block RAM is consumed much more than other resources. The most-consumed resource becomes the bottleneck of throughput enhancement as in the wooden barrel theory. Some designs choose to modify the original compression format to facilitate hardware design [15]. In addition, most existing studies only focus on designing acceleration kernels but ignore supporting software. These drawbacks will directly degrade the design practicability. Moreover, in practical applications, various scenarios may demand different algorithms, compression performance, resource usage, processing modes, etc.

To overcome the above challenges, in this paper, we propose **HybriDC**, an adaptive resource-efficient CPU-FPGA heterogeneous acceleration system for lossless data compression. HybriDC provides a universal end-to-end compression acceleration framework for various compression methods. In HybriDC, an algorithm fine-tuning method based on performance–resource modeling is proposed to optimize the hardware compression kernel design. To improve the resource efficiency of compression kernels, a novel memory-efficient parallel hash table with an extra checksum is proposed in HybriDC. In particular, we develop an end-to-end compression system prototype of HybriDC based on the LZ4 compression algorithm [18]. The design goal of this system prototype is to achieve the highest throughput with the given available resources, while ensuring the compression ratio is better than that of compression software. The HybriDC system is adaptable to most compression algorithms and not limited to LZ4. In summary, our contributions include the following:We present a universal end-to-end compression acceleration framework in HybriDC. It defines how to assign the compression workloads to the CPU-FPGA heterogeneous platform. The proposed heterogeneous architecture ensures application compatibility and performance scalability.By a hardware–software codesign, we build a performance–resource model of the compression algorithm concerning the design goal, compression performance, hardware resources, etc. Based on the model, we deduce the resource-balanced design principle and fine-tune the compression kernel parameters. As a result, the block RAM usage of the LZ4 kernel is reduced by 32%.According to the resource-balanced principle, we use less-consumed logic resources to increase the degree of parallelism of the compression kernel. In particular, we propose a memory-efficient parallel hash table enhanced with a checksum, which can improve the bandwidth and compression ratio without extra memory overhead.We develop an end-to-end compression system prototype of HybriDC and evaluate it in detail. Compared to previous designs with comparable compression ratios, our compression kernel achieves a 2.5–4× improvement in memory efficiency. Based on balanced resource utilization, the system prototype obtains a good performance scalability. The four-kernel system prototype achieves a threefold increase in power efficiency over standard software.

The remainder of this paper is organized as follows. Section 2 presents the heterogeneous compression acceleration framework of HybriDC. Section 3 introduces the algorithm fine-tuning method based on performance–resource modeling. The hardware compression kernel design of HybriDC is exhibited in Section 4. The evaluation results of HybriDC are reported in Section 5. Section 6 reviews related work. Our conclusions are drawn in Section 7.

## 2. Heterogeneous Compression Acceleration Framework

The heterogeneous compression acceleration framework of HybriDC defines how to assign the end-to-end compression task to the CPU-FPGA heterogeneous platform. In addition, it provides a scalable heterogeneous architecture for high-performance data compression.

### 2.1. Compression Task Assignment on Heterogeneous Platforms

Different parts of heterogeneous platforms are good at different workloads. The reasonable compression task assignment can improve the resource efficiency of heterogeneous platforms. This subsection analyzes the end-to-end lossless data compression workflow and explains how to assign various compression workloads based on heterogeneous hardware complementarity.

#### 2.1.1. End-to-End Lossless Data Compression Overview

As shown in Figure 1, most lossless data compression methods have the same essential workflow consisting of three stages. First, raw data are divided into fixed-size blocks, which can facilitate parallel computing. Then, raw blocks are compressed into compressed blocks by the compression kernel. The compressed block consists of two parts: the block header, which records block information such as the compressed block size and the compressed segments, which are the basic compressed data. Finally, these compressed blocks are packed in order. A compression header is attached in front of the compressed data, which contains some key information, such as the compression type and the total compressed size.

#### 2.1.2. Workloads Assignment based on Hardware Complementarity

Different parts of the CPU-FPGA heterogeneous platform are highly complementary. FPGAs contain plenty of programmable computational resources, which can be customized into high-performance processing units. However, due to the limited on-chip memory size (usually less than 100 MB), it is hard for FPGAs to implement functions requiring much memory space. Instead, CPUs are equipped with abundant memory, while they are not good at parallel fine-grained processing.

Based on hardware complementarity, various workloads of end-to-end data compression are assigned to the CPU part and the FPGA part in HybriDC. Specifically, the block compression workload is computing-intensive and parallelizable. Thus, the FPGA is responsible for offloading the block compression workload. In addition, dividing raw data and packing compressed blocks require significant memory space for buffering the raw and compressed data. Therefore, these two workloads are assigned to the CPU part in HybriDC.

### 2.2. Scalable Heterogeneous Architecture for High-Performance Data Compression

By a software–hardware codesign, the heterogeneous architecture of HybriDC can promise application compatibility and performance scalability. As displayed in Figure 2, the FPGA part and the CPU part are connected by the PCIe interface, which is compatible with commodity servers. The input data are read from the host memory through PCIe with direct memory access (DMA). Similarly, the output data are directly written to the host memory via DMA after being compressed.

#### 2.2.1. Software Design

The software program is a vital part of HybriDC. On the one hand, as presented in Section 2.1.2, software flexibility is an excellent complement to hardware. Specifically, a data decollator is designed to divide the large-size data into fixed-size blocks. In addition, a data packer is designed for packing compressed blocks. Importantly, blocks of different tasks may be out-of-order after hardware processing. To address the challenge, a unique sequence number is allocated to each block, which will not change in hardware processing.

On the other hand, the HybriDC software architecture can offer compatibility, which is significant for practical applications. To stay compatible with standard compression software, the acceleration API is designed like the original software API. The acceleration API can be called directly by advanced applications. Moreover, to facilitate high-performance parallel computing at the software level, the software system is designed to support asynchronous processing and multithreading, two common data processing modes in high-performance scenarios.

#### 2.2.2. Hardware Design

As the core of the HybriDC system, the hardware part aims to accelerate the block compression workload. As shown in Figure 2, the HybriDC hardware architecture can support scalable multiway parallel block compression.

*Compression kernels* are the cornerstone of HybriDC hardware. In HybriDC, multiple compression kernels can execute concurrently to achieve scalable throughput. The maximum number of compression kernels is determined by available hardware resources and resource efficiency. Thus, the resource efficiency of HybriDC is important to performance scalability. The detailed design of the compression kernel is presented in Section 4.

Except for the necessary compression kernel, several components are designed to support parallel processing of compression kernels, including data schedulers, data reshapers, and data buffers.

*Data schedulers*, including an input scheduler and an output scheduler, are designed to schedule I/O data between the CPU software and compression kernels. Specifically, for each raw block from the CPU, the input scheduler allocates it to an idle compression kernel. In addition, the output scheduler is responsible for aggregating compressed blocks from different compression kernels. A round-robin scheduling strategy is used for simplicity and efficiency. Other schedule strategies can also be applied according to different requirements.

*Data reshapers* are designed to adjust the data flow width. In the hardware design, the data flow width of the DMA engine differs from that of compression kernels. Through the data reshaper, the input data width is reshaped to the data width of the output module.

*Data buffers* can make the HybriDC system smoother. On the one hand, when the processing pipeline is stalled in some steps, the data buffer is used for storing the temporary data and adjusting the data flow. On the other hand, the clock frequency of the DMA usually differs from that of compression kernels. Thus, data buffers based on asynchronous FIFOs are applied to transfer data across clock domains.

## 3. Algorithm Fine-Tuning Based on Performance–Resource Modeling

In this section, we review the target compression algorithm first. Then, we explain how to build the performance–resource model of the compression algorithm. Finally, we present the algorithm fine-tuning method. Note that the modeling and optimization method can also be used for other compression algorithms.

### 3.1. LZ4 Compression Algorithm Review

We choose to base our prototype design on LZ4 as it has been widely used in many scenarios, such as big data processing and embedded computing [19]. As a variant of the LZ77 algorithm [20], LZ4 is a dictionary-based compression algorithm [21]. LZ4 realizes data compression by replacing the repeated data with the index of the same past data. It consists of five major phases, including *Hash Calculation*, *Hash Table Update*, *First Match*, *Extended Match*, and *Sequence Encoding* (Figure 3).

*Hash Calculation*. In LZ4, data are processed byte by byte. The current 4-byte string, consisting of the current byte and the following 3 bytes, is considered as the basic input to be matched. In *Hash Calculation*, the hash value of the 4-byte string is calculated by a hash function.

*Hash Table Update*. LZ4 uses the hash table to look up the repeated data in O(1) time complexity. As presented in Algorithm 1, indexed by the hash value of the current string, the candidate address of the possible matched past string is fetched from the hash table. After that, the address of the current string is stored, substituting the previous one.
**Algorithm 1:** Hash table update algorithm**Input:** current_data: the current data to be matched**Output:** candidate_address: the candidate address of past data   // **Hash value calculation**   hash_value←HashFunction(current_data)   // **Hash table lookup**   candidate_address←hash_table[hash_value]   // **Hash table insert**   hash_table[hash_value]←AddressOf(current_data)

*First Match*. A possible hash collision (different data maps to the same hash value) causes a false-positive lookup of the hash table. Thus, in *First Match*, the specific past string is read from the past data buffer (called a dictionary) to confirm the match result. If the past string and the current string match successfully, *Extended Match* is executed to find a longer match. Otherwise, the input window slides forward one byte, and the above procedures repeat.

*Extended Match*. Through *First Match*, a preliminary 4-byte match is found. In *Extended Match*, bytes following the matched current string and past string are compared to confirm the maximum match length. After this phase, one round of match searching finishes.

*Sequence Encoding*. After each round of match searching, a 5-tuple consisting of the unmatched string, the unmatched length, the matched string, the matched length, and the address offset between matched current and past strings is obtained. The 5-tuple is encoded into an LZ4 sequence, which is the basic compressed segment of the LZ4 compression block. Specifically, the matched string is substituted with the matched length and the address offset.

To conclude, the LZ4 algorithm stores the addresses of the most recent strings in the hash table. The matched strings are found via the hash table quickly. After *First Match* and *Extended Match*, repeated data are found and the input data are encoded as LZ4 sequences. The sequential attribute of the LZ4 compression algorithm makes it suitable for an hardware implementation on the FPGA.

### 3.2. Compression Performance Modeling

Original compression algorithm parameters designed for software may not fit the hardware implementation. Thus, a performance–resource model was built to guide the compression kernel design. It concerned the compression ratio, throughput, available hardware resources, the design goal, etc.

#### 3.2.1. Compression Ratio

The compression ratio is calculated by $\frac{uncompressed\;size}{compressed\;size}$. In LZ77-type compression algorithms, the compression ratio performance is determined by the capability of searching the repeated data. In LZ4, four parameters are mainly related to the search capability, including the hash function, the hash table size, the dictionary size, and the maximum literal size (Table 2a).

For the hash table, hash and $S_{HT}$ influence the probability of a hash collision. As introduced in Section 3.1, more hash collisions cause more false-positive lookups and further decrease the compression ratio. The dictionary stores the most recent input data, whose size ($S_{DICT}$) constrains the match range. In *First Match*, the processed unmatched literals should be stored in a literal buffer until a match is found. $S_{ML}$ indicates the maximum size of the literal buffer. The compression ratio determined by these four parameters is denoted as $CR(S_{DICT},S_{ML},S_{HT},hash)$.

#### 3.2.2. Throughput

In the default LZ4, the input window slides forward one byte in each round of *First Match*. To enhance the throughput, the larger slide stride can be set in LZ4 software. However, the compression ratio will significantly decline since bytes between the two input windows will be ignored when searching match.

In hardware compression kernel designs, bytes between two input sliding windows can be processed concurrently [16,22]. Thus, the compression ratio does not lose much by using parallel processing in the hardware compression kernel. *P* denotes the degree of intrakernel parallelism, the average bytes processed per cycle in the compression kernel. The compression kernel throughput ($T_{kernel}$) is calculated by P·f, where *f* means the design frequency. The theoretical end-to-end throughput of the HybriDC system with *N* compression kernels can achieve
(1)$$T_{sum}=N\cdot P\cdot f.$$

#### 3.2.3. Hardware Resource Usage

The hardware compression kernel consumes various on-chip resources (Table 2c), including LUTs, FFs, block RAMs, etc.

LUTs are mainly used to construct the combinational logic circuit. FFs can buffer data in the clocked sequential circuit design. In the compression kernel design, increasing *P* notably complexifies the hardware design and raises the LUT usage ($LUT_{kernel}$) and the FF usage ($FF_{kernel}$). We denote $LUT_{kernel}$ and $FF_{kernel}$ as functions of *P*:(2)$$\left\{\begin{array}{c} LUT_{kernel}=LUT\left(P\right),\\ FF_{kernel}=FF\left(P\right). \end{array}\right.$$

Block RAMs are mainly used in implementing large buffers, including the dictionary, hash table, and literal buffer. The block RAM usage can be estimated by related components’ sizes:(3)$$RAM_{kernel}=S_{DICT}+S_{ML}+S_{HT}.$$

Consequently, *N* compression kernels should consume $N\cdot LUT_{kernel}$ LUTs, $N\cdot FF_{kernel}$ FFs, and $N\cdot RAM_{kernel}$ block RAMs.

#### 3.2.4. Optimization Model Formulation

For the HybriDC system prototype, its design goal was to maximize the compression throughput. Design variables included *N*, *P*, $S_{DICT}$, $S_{ML}$, $S_{HT}$, and hash. Moreover, there were two design constraints. First, resources consumed by *N* compression kernels could not exceed the available resources (Table 2c). Second, the compression ratio of HybriDC could not be less than the target value ($CR_{0}$).

According to Equations (Equation 1)–(Equation 3), HybriDC’s performance optimization model can be obtained as:(4)$$\underset{N,P,S_{DICT},S_{ML},S_{HT},hash}{\mathrm{argmax}}\left(N\cdot P\cdot f\right),$$
subject to
(5)$$\left\{\begin{array}{c} CR\left(S_{DICT},S_{ML},S_{HT},hash\right)⩾CR_{0},\\ N\cdot LUT\left(P\right)⩽LUT_{avl},\\ N\cdot FF\left(P\right)⩽FF_{avl},\\ N\cdot \left(S_{DICT}+S_{ML}+S_{HT}\right)⩽RAM_{avl}. \end{array}\right.$$

The algorithm parameters fine-tuning strategy was developed based on the performance optimization model.

### 3.3. Algorithm Parameters Fine-Tuning

#### 3.3.1. Resource-Balanced Design Principle

Increasing *N* and *P* can improve throughput (Equation (Equation 1)) but increases resource consumption (Equations (Equation 2) and (Equation 5)). For *N* compression kernels, the usage of each resource cannot exceed its total available amount (Equation (Equation 5)). Thus, the most-consumed resource decides the maximum *N* ($N_{max}$). In other words, $N_{max}$ depends on the resource with the largest consumption proportion, that is
(6)$$N_{max}=\left\lfloor \frac{1}{\max(PCT_{LUT},PCT_{FF},PCT_{RAM})}\right\rfloor ,$$
where PCT means the resource percentage of one kernel consumption to the total available amount. Consequently, we derived the resource-balanced design principle from Equation (Equation 6) to improve throughput:(1)*Minimizing the usage of the most-consumed resource;*(2)*Using the less-consumed resources to improve performance.*

As presented in Table 1, the consumption of block RAMs is much higher than that of other resources in existing designs. According to Principle (1), we fine-tuned the algorithm parameters to decrease the consumption of block RAMs in the HybriDC system design. Moreover, increasing *P* increases the logic resource consumption but does not consume more block RAMs (Section 3.2.3, Equation (Equation 2)). Therefore, according to Principle (2), we increased *P* by the less-consumed LUTs and FFs in the compression kernel design.

#### 3.3.2. Parameters Fine-Tuning with Compression Ratio Constraint

Memory-related parameters ($S_{DICT},S_{ML},S_{HT}$) significantly affect the compression ratio performance. When fine-tuning these parameters, the compression ratio constraint must be taken into account (Equation (Equation 5)).

To analyze the impact of changing parameters, we adjusted each parameter individually and observed the compression ratio changes by a software simulation. For example, when analyzing $S_{HT}$, the value of $S_{HT}$ changed from 2 KB to 64 KB while the other parameters kept their default values. The default values of ($S_{DICT},S_{ML},S_{HT}$) were (64 KB, 64 KB, 16 KB). The results tested with the Silesia compression corpus [23] are illustrated in Figure 4. The compression ratio increased with $S_{DICT}$ and $S_{HT}$, while the growth trend slowed down. Differently, the compression ratio barely changed when $S_{ML}$ exceeded 2 KB, since a large-size literal was rare in most data.

Based on the experimental results, we could reasonably fine-tune the algorithm parameters. To save block RAMs, $S_{ML}$ was set to 2 KB in the compression kernel. A 2 KB buffer consumes just one block RAM in common FPGAs (one Intel M20K BlockRAM or one Xilinx 18K BRAM). Considering the compression ratio constraint, $S_{DICT}$ was set to 64 KB, the default size in LZ4 software. The compression ratio may slightly decline due to the parallelization of the compression kernel. Thus, we increased $S_{HT}$ to 32 KB, compensating for the compression ratio loss.

In addition to memory-related parameters, the hash function also affects the compression ratio. We analyzed two hash function candidates, including the Fibonacci hash and the shift-and-xor (SAX) hash. The Fibonacci hash is the default hash function in LZ4. It generates the hash value by selecting partial bits from the product of the 32-bit input string and a 32-bit Fibonacci number. The SAX hash performs two bitwise operations, SHIFT and XOR, on the input data to obtain the hash value [24]. According to our evaluation, the Fibonacci hash (the standard LZ4) performed better than the SAX hash (Figure 4). Thus, the Fibonacci hash was chosen in the compression kernel design.

In conclusion, based on the performance–resource model, the algorithm parameters were fine-tuned for the hardware design. When meeting the compression ratio constraint, the block RAM usage of the LZ4 compression kernel could be reduced from 64+64+16=144 KB to 64+32+2=98 KB (32% decrease). The HybriDC design thus became more resource-efficient. Due to less block RAM usage, the maximum kernel number increased (Equation (Equation 6)). Thus, the HybriDC design could achieve a higher throughput (Equation (Equation 1)).

## 4. Hardware Compression Kernel Design

In HybriDC, the compression kernel was implemented to support intrakernel parallel processing. According to the resource-balanced design principle, LUTs and FFs are used more and the block RAM usage does not increase. In this section, we propose the detailed architecture of the parallel compression kernel (Figure 5). Furthermore, we introduce the memory-efficient parallel hash table design, which is the core module of the compression kernel.

### 4.1. Compression Kernel Architecture

The HybriDC compression kernel consists of several modules. All modules are tailored to support multibyte parallel processing.

*A. The input parallelization module*. The input is parallelized to *P* input strings in this module. As shown in the top left corner of Figure 5, *P* input strings are constructed from P+3 bytes. In each new cycle, the processing window slides forward *P* bytes, and the new *P* input strings are generated.

*B. The hashing module*. In each cycle, the hashing module receives *P* strings from the input parallelization module and outputs their hash values to the next module. In this module, DSPs are used to implement the *P*-way Fibonacci hash calculation.

*C. The hash table update module*. In this module, a parallel hash table supporting updating *P* records concurrently was implemented. For *P* hash values, *P* corresponding dictionary addresses are output in each cycle. In addition, for each record in the hash table, a checksum was implemented by idle block RAM space. The extra checksum reduces the number of false-positive lookups and improves the compression ratio indirectly. The details of the parallel hash table design are explained in Section 4.2.

*D. The dictionary*. As mentioned in Section 3.2, the dictionary stores the most recent input data. When receiving the read requests with addresses from two match modules, the dictionary returns the corresponding past input data. To avoid a possible access collision, the dictionary processes one read request in each cycle.

*E. The first match module*. The current and past strings are compared to confirm the match result in this module. Due to the read request limitation, just one candidate address can be processed in the dictionary. To find the longest match, the earliest candidate (the input string near the beginning) among *P* possible match candidates is selected. The *P*-byte past string reads from the dictionary and the *P*-byte current string are compared by the longest prefix match. Supposing the match length is *l* bytes, there are three possible match results, defined as *failed match* (l<4), *partial match* (4⩽l<P), and *complete match* (l=P).

A *failed match* means this *P*-byte input string is unmatched. A *partial match* means that a matched string is found, but the last several bytes of the input string are unmatched. This match round is over with a *partial match*. A *complete match* means that a longer matched string can be found by the extended match.

*F. The extended match module*. When the match result in the last cycle is a *complete match*, the extended match module is used. Similarly, the *P*-byte past string reads from the dictionary are compared with the current string by the longest prefix match. The results can be classified into *partial match* (l<P) and *complete match* (l=P). When obtaining a *complete match*, the extended match will work continuously in next cycle. If the result is a *partial match*, this match round finishes.

*G. The output encoding module*. Once a match round has finished, a set of compression information is generated, including the unmatched string, the unmatched length, and match information (length and offset). The output encoding module encodes the information into *sequences*. Specifically, different pieces of information are processed in order. Thus, a finite state machine (FSM) was designed to control these steps in this module.

*H. The match-refining mechanism*. As mentioned in Section 3.2.2, some data may miss the matching process caused by the sliding input window. For example, assume that in an 8-byte processing window “abcdefgh”, the substring “abcd” is matched. If the processing window slides forward 8 bytes in the next cycle, the string “efgh” will be skipped and considered as an unmatched string. Nevertheless, the unmatched bytes “efgh” may be matched with other past strings. To address the problem, we propose a match-refining mechanism for finer-grained processing. If the last match result was a *partial match*, the processing window begins after the last matched byte. In the previous example, the next 8-byte processing window will follow “abcd” and should be “efghxxxx”. Thanks to the refining mechanism, the compression ratio performance improves significantly. Naturally, the actual degree of intrakernel parallelism of the compression kernel is less than *P* since the sliding step is less than *P*.

### 4.2. Memory-Efficient Parallel Hash Table Design

As mentioned before, the hash table is implemented by block RAMs. In general, a block RAM has two access ports for reading and writing data. Multiple block RAMs can be combined into a large storage component (Figure 6a). However, the number of access ports does not increase for maintaining the data consistency. Thus, this combination of block RAMs cannot realize a parallel hash table.

Several studies applied the LVT-based multiport memory scheme to implement a parallel hash table [16,22] (Figure 6b). Nevertheless, this scheme consumes many memory resources [25]. For instance, to construct a 16 KB hash table with *P* reading ports and *M* writing ports, P×M×16 KB block RAMs should be used. Obviously, this heavy use of block RAMs causes a worse imbalanced resource usage.

In HybriDC, we propose a memory-efficient parallel hash table design, which does not cause extra memory usage. In particular, we use the idle space of block RAMs to implement a novel checksum design and thus improve the compression ratio.

#### 4.2.1. Multiport Hash Table Implementation

The hash table architecture in HybriDC is described in the middle bottom of Figure 5. In the HybriDC hash table, each block RAM is instantiated individually, and each access port can be used concurrently (Figure 6c). Meanwhile, an access forwarding component is designed to forward multiple access requests to the corresponding block RAMs.

An example of parallel multiple access in our design is shown in Figure 7a. Four access requests (R0, R1, R2, R3) in channels (C0, C1, C2, C3) are targeted at hash table addresses (4, 3, 7, 0). Addresses (4, 3, 7, 0) are located in block RAMs (B2, B1, B3, B0). In this case, R0, R1, R2, and R3 are forwarded to B2, B1, B3, and B0, respectively. Then, B0, B1, B2, and B3 read and update contents at addresses 0, 3, 4, and 7. Finally, results output from addresses 0, 3, 4, and 7 are sent back to the corresponding channels C3, C1, C0, and C2.

However, some requests may be targeted on the same block RAM. In this situation, only one request can be processed (supposing that one block RAM can handle only one request). As mentioned before, choosing the request of the earliest input string is helpful to find a longer match. Therefore, the earliest first selection strategy is applied in the forwarding component to alleviate the effects of access collision. As shown in Figure 7b, R0 and R1 target the same block RAM B1. The earliest request R0 is processed, and R1 is abandoned.

#### 4.2.2. Checksum Design for Lookup Optimization

The hash table designs of existing studies can be classified into two types: storing dictionary addresses with or without input strings. Both types waste much storage capacity due to inadequate block RAM utilization.

*Type 1: the hash table stores only dictionary addresses.* For the block RAM storage, its width and depth are fixed. Take the M20K block RAM in Intel FPGAs as an example, its width is 20 bits, and its depth is 1024. For a 64 KB dictionary, $\log_{2}\left(64\times 2^{10}\right)=16$ bits addresses are used as the index and stored in the hash table. Obviously, the 16-bit dictionary address cannot fill the 20-bit width of block RAMs, i.e., (20−16)/20=20% storage capacity is wasted.

*Type 2: the hash table stores dictionary addresses and input strings.* The hash table lookup may be a false positive due to the hash collision. Some studies proposed storing the 4-byte input string with its address in the hash table. By skipping reading data from the dictionary, this scheme can facilitate the match result verification in the first match phase. In this scheme, the 48-bit hash table records (16-bit addresses and 32-bit strings) consumes triple block RAMs. In this case, (60−48)/60=20% storage capacity is wasted.

In HybriDC, we propose a checksum design utilizing the spare space of block RAMs to filter the false-positive lookup. As shown in Figure 5, a checksum calculation component is implemented in the hash table update module, which generates a checksum for each input string. The checksum size equals the width of the spare space of block RAMs. A checksum and a dictionary address are regarded as a record to be stored in the hash table. Once a hash table record has been looked up, the new checksum is compared with the fetched checksum. The failed matching result of the checksum comparison means that this lookup must be a false positive. Therefore, most false-positive lookups can be filtered by the checksum, which improves the compression ratio performance indirectly.

## 5. Evaluation

In this section, we evaluate the performance of HybriDC. As most existing studies just implemented the compression kernel, we compared the proposed compression kernel to related studies. Furthermore, we evaluated various aspects of the HybriDC system prototype.

### 5.1. Compression Kernel Evaluation

In this subsection, we evaluate the performance of our compression kernel and exhibit its improvement over other designs. The evaluation criteria include compression throughput, compression ratio, resource utilization, resource efficiency, etc.

#### 5.1.1. Experimental Setup

We implemented the hardware part of the HybriDC system on a customized acceleration card with an Intel Arria 10 FPGA chip (10AX048H2F34E2SG). The FPGA development tool was Intel Quartus Prime Professional Edition 18.0. The hardware description language we used was Verilog. We used ModelSim to simulate the compression process and obtain accurate processing cycles. Then, we multiplied the number of cycles by the clock frequency to calculate the compression throughput. The Calgary corpus [26] was used in the compression kernel evaluation. The HybriDC compression kernel performance is presented in Table 3. Table 4 exhibits the detailed comparison between HybriDC and other LZ4 accelerators.

#### 5.1.2. Compression Ratio

As shown in Table 3, the overall compression ratio of HybriDC was 1.951, which was higher than 1.928 for the LZ4 algorithm and achieved our compression ratio goal. The improvement of the compression ratio derived from the hash table size fine-tuning and the checksum design. Due to data redundancy, the compression ratios were quite different in various data classes. For example, image data usually contain lots of consecutive identical values, i.e., redundant data. Thus, the compression ratio of *pic* was the best in the Calgary corpus.

#### 5.1.3. Compression Throughput

In our prototype, the degree of intrakernel parallelism *P* was set to eight. The clock frequency of the compression kernel was 100 MHz. The theoretical throughput could achieve up to 8×100=800 MB/s. The actual throughput performance was different on various data because of the match-refining mechanism (Table 3). As mentioned in Section 4.1, once a *partial match* happens, the processing sliding window should slow down to check the unmatched data of the last cycle. Thus, the throughput decreases with the frequency of *partial matches*. Instead, the throughput increases if *failed matches* and *complete matches* happen more frequently. For instance, geo had a low compression ratio but a high throughput, since most match results were *failed matches* when compressing geo. Similarly, pic with a high compression ratio had the highest throughput due to massive *complete matches*.

#### 5.1.4. Compression Performance Comparison

Compared to other designs, our design achieved the highest throughput with no reduction in the compression ratio. As shown in Table 4, the average throughput of the HybriDC kernel was up to 562.88 MB/s, which was better than the previous designs with comparable compression ratios. With the advantage of design frequency, Benes’ scheme obtained a higher throughput than that of the HybriDC kernel. However, its compression ratio performance was reduced a lot because of its small hash table size. In many scenarios (such as data storage), a better compression ratio can decrease the data size more, which is more important than throughput performance. HybriDC could provide a high throughput and maintain the compression ratio performance. In conclusion, the compression performance of HybriDC was more balanced and desirable.

#### 5.1.5. Resource Utilization Efficiency

As mentioned before, performance scalability benefits from balanced resource utilization. Since the block RAM is the most-consumed resource, memory efficiency was used to measure resource utilization efficiency. Its value was calculated by $\frac{block\;RAM\;size}{throughput}$.

As shown in Table 4, HybriDC achieved state-of-the-art memory efficiency while maintaining an adequate compression ratio. In other words, given the same memory resources, HybriDC could obtain the highest throughput. Compared to MLZ4C and Xilinx, two designs with comparable compression ratios, the memory efficiency of HybriDC was 4× and 2.5× better, respectively. Based on its good memory efficiency and balanced resource utilization, the HybriDC system could integrate more compression kernels and gain higher throughput performance.

### 5.2. HybriDC System Performance

The HybriDC system prototype integrates multiple compression kernels to provide more throughput. In this subsection, we report on the performance of this system prototype. The end-to-end throughput, resource utilization, and power efficiency are evaluated. The multikernel scalability of HybriDC is discussed primarily.

#### 5.2.1. Experimental Setup

The HybriDC prototype consisted of the FPGA card and the CPU host. In the FPGA part, multiple compression kernels were integrated. The host was equipped with an Intel Core i5-7500 CPU @ 3.4 GHz with 6 MB of L3 cache and 4GB memory, running CentOS Linux 7 with kernel 3.10. The FPGA card and the host connected by the PCIe 3.0 x4 interface. We use three standard compression benchmark datasets to evaluate the HybriDC system, including the Calgary corpus, the Canterbury corpus [27], and the Silesia corpus.

#### 5.2.2. End-to-End System Throughput

We evaluated the end-to-end throughput of the HybriDC system in the asynchronous processing mode. Specifically, test data were continuously sent from the host to the accelerator for compression. When data sending began and data receiving ended, the precise time was recorded on the host side. Then, the end-to-end throughput was calculated with the processing time and the data size. We changed the number of compression kernels from one to four to evaluate the performance scalability of HybriDC.

As shown in Figure 8, with four kernels, the maximum end-to-end throughput of HybriDC could achieve about 2.7 GB/s (ptt5). It is worth noting that the end-to-end throughput of the HybriDC system increased linearly with the number of kernels. In other words, with enough hardware resources, the throughput of the HybriDC system could be expanded easily by increasing the number of compression kernels.

#### 5.2.3. FPGA Resource Utilization

The FPGA resource utilization means much to the performance scalability of HybriDC. As mentioned in Section 2.2.2, except for the compression kernels, peripheral hardware components were implemented in the HybriDC hardware system and consumed some extra resources. The DMA data mover was not related to the number of kernels, whose resource consumption was nearly constant. Table 5 shows the FPGA resource utilization comparison between the single compression kernel and the entire system prototype integrated with four kernels (excluding the DMA data mover). The main consumed extra resource was the block RAM, which was mainly used by the peripheral data buffer. As a whole, the additional resource consumption was acceptable for scaling throughput performance. The four-kernel system prototype consumed less than 50% of the FPGA resources. As indicated in our further experiments, the HybriDC system could integrate at least eight compression kernels on the evaluated FPGA chip.

#### 5.2.4. Power Efficiency

Power efficiency is a vital performance requirement in many application scenarios, including data centers, mobile computing, etc. Compared with LZ4 software, HybriDC required some additional FPGA power but less CPU power. Importantly, HybriDC brought a significant performance improvement. As shown in Table 6, the FPGA power was only 6.14 W while HybriDC could provide more than three times the throughput of LZ4 software. As a result, the four-kernel HybriDC prototype achieved about three times more power efficiency than LZ4 software.

In addition, we could further improve the power efficiency of HybriDC by integrating more kernels. In practice, the overall power efficiency increases with FPGA resource utilization. The HybriDC system integrated with eight kernels is expected to realize a power efficiency four times greater than that of LZ4 software.

## 6. Related Work

### 6.1. LZ4 Acceleration

The LZ4 algorithm was proposed by Yann Collet in 2011. After that, several works have reported on the FPGA-based LZ4 compression acceleration [11,14,15,16,22]. In 2015, Bartik et al. first tried using an FPGA to implement LZ4 for 4K transmission compression [14]. This design could process one byte per cycle. The design reduced the hash table size from 4096 records to 1024 records to save memory resources. However, this modification significantly degraded its compression ratio. In 2019, Bartik et al. improved their work and proposed an optimized match search unit (MSU) to support parallel processing [16,22]. Nevertheless, its memory cost increased exponentially with the number of parallel access ports [25]. To improve the design clock frequency, Liu et al. modified the original LZ4 algorithm and proposed MLZ4 [15]. Their design relied on a new compression format to reduce the literal buffer size and stabilize the output delay. The modified compression format was a double-edged sword since it was incompatible with the original LZ4 software program. High-level synthesis (HLS) is an emerging technology that enables the use of high-level programming languages (e.g., C/C++, Python, et al.) to perform hardware design [28,29]. Xilinx used its HLS tool to implement the LZ4 acceleration [11]. However, the lack of elaborate optimization made this design cost significant memory resources. In addition, its acceleration kernel throughput was just one byte per cycle without intrakernel parallelism.

In summary, existing studies come short in terms of memory efficiency and compatibility. By comparison, the HybriDC system provides a compatible heterogeneous compression acceleration framework and realizes a good memory-efficient compression kernel.

### 6.2. Other Lossless Data Compression Acceleration

Besides LZ4, many studies have tried to accelerate other lossless data compression methods, such as gzip, Xpress9, Deflate, and bzip2 [17,24,30,31]. Abdelfattah et al. proposed using the OpenCL HLS tool to design a gzip accelerator [24]. Via OpenCL, this work was completed in just one month. However, its block RAM usage was significantly more than that of the Verilog-based competition. Kim et al. presented a heterogeneous Xpress9 compressor with scalable performance under a heavily multithreaded environment [30]. Since the complex match selection logic of Xpress9 hindered the pipeline, the compressor’s throughput was relatively low. In 2020, Ledwon et al. applied the Xilinx HLS tool to implement a high-throughput Deflate compressor [17]. Due to the low data dependency of Deflate, this design achieved a throughput of 4 GB/s with a 250 MHz clock frequency and a degree of intrakernel parallelism of 16. Qiao et al. designed a Burrows–Wheeler transform (BWT) accelerator for bzip2 [31]. It supported a compression block size of up to 500 KB, which could bring an outstanding compression ratio performance. Nevertheless, similar to the bzip2 software program, its throughput was relatively low.

In conclusion, these designs based on various algorithms and tools achieved different compression performance values. As an adaptive compression acceleration solution, HybriDC can be used to improve these designs further.

## 7. Conclusions

Lossless data compression is widely applied in various data-centric scenarios while bringing a huge CPU overhead. In this paper, HybriDC, an adaptive resource-efficient CPU-FPGA heterogeneous acceleration system was proposed to accelerate compression workloads. HybriDC provides a compatible end-to-end compression acceleration framework. The proposed LZ4 compression kernel demonstrated a state-of-the-art memory efficiency at a comparable compression ratio. The results of the prototype evaluation showed the desirable performance scalability of HybriDC. Moreover, HybriDC obtained much better power efficiency than other compression software.

There are several opportunities to improve this work in the future. Since the match-refining mechanism brings data dependency, the clock frequency of the current design is limited. Therefore, optimizing the match-refining mechanism can improve the clock frequency further. Moreover, the block RAM is still the most-consumed resource in our prototype design. It is worth increasing the degree of intrakernel parallelism by using the less-consumed resources.

## Figures and Tables

**Figure 1 micromachines-13-02029-f001:**
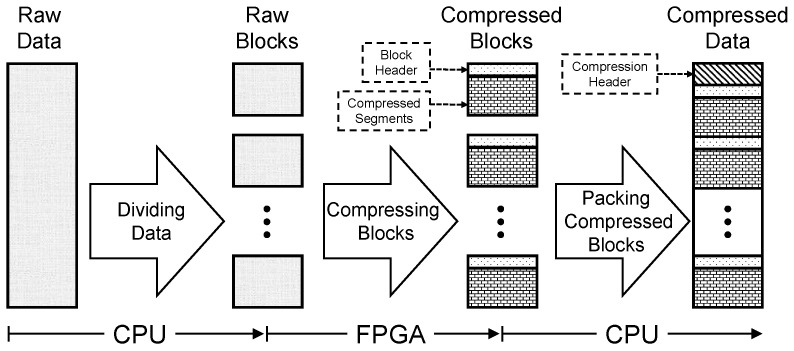
The essential workflow of end-to-end lossless data compression, and the heterogeneous workloads’ assignment strategy of HybriDC.

**Figure 2 micromachines-13-02029-f002:**
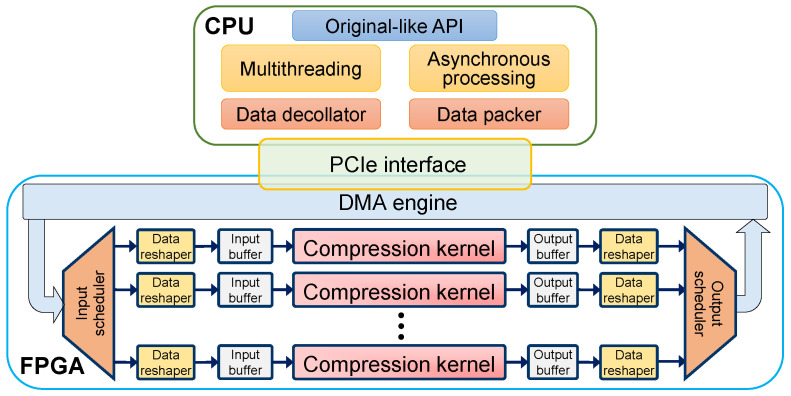
The architecture of the HybriDC system based on the CPU-FPGA heterogeneous platform.

**Figure 3 micromachines-13-02029-f003:**
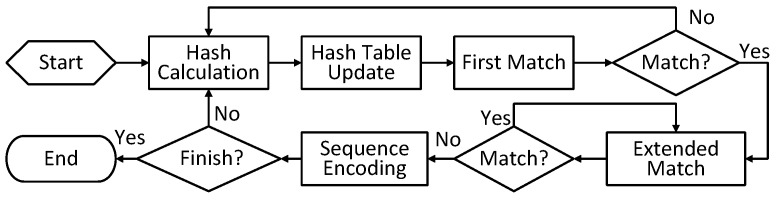
The LZ4 workflow.

**Figure 4 micromachines-13-02029-f004:**
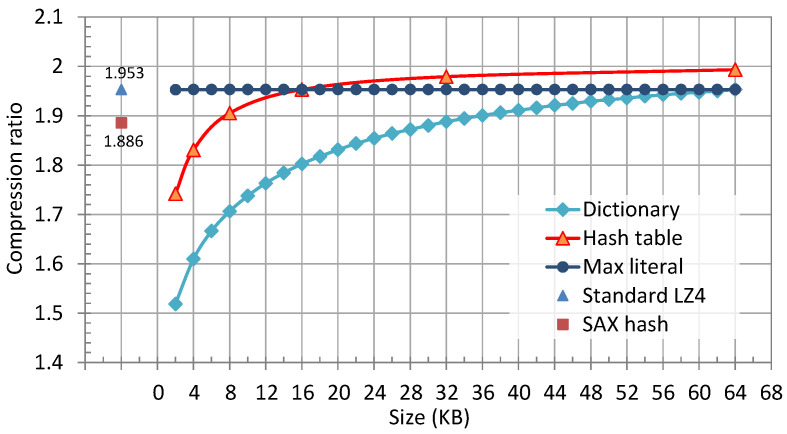
The compression ratio performance with different algorithm parameters.

**Figure 5 micromachines-13-02029-f005:**
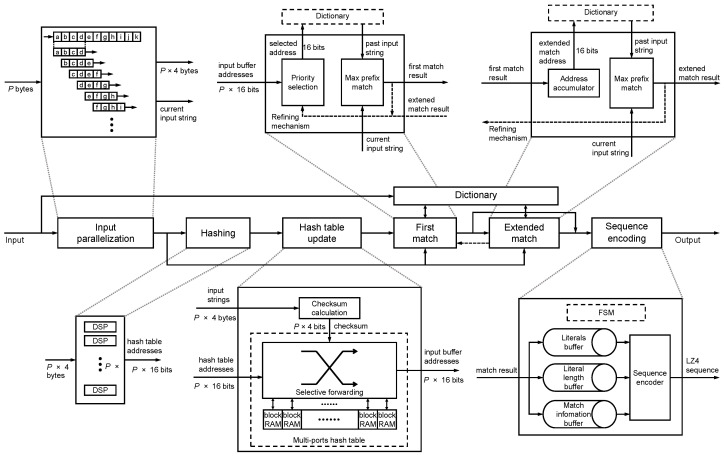
The workflow of the HybriDC compression kernel and the specific architecture of each module.

**Figure 6 micromachines-13-02029-f006:**
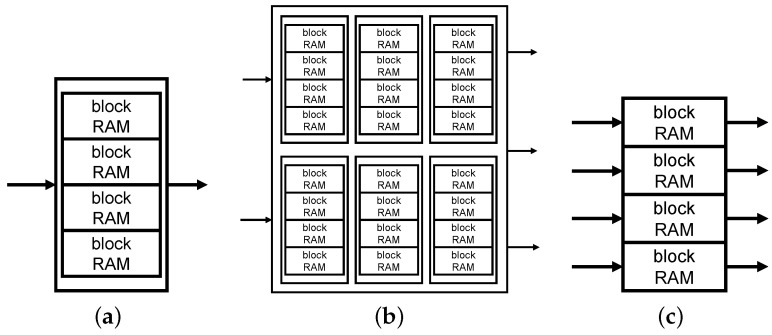
Block RAM usage patterns of different hash tables. (**a**) Normal combination (1 writing port and 1 reading port), (**b**) LVT-based hash table (2 writing ports and 3 reading ports), (**c**) HybriDC hash table (4 writing ports and 4 reading ports).

**Figure 7 micromachines-13-02029-f007:**
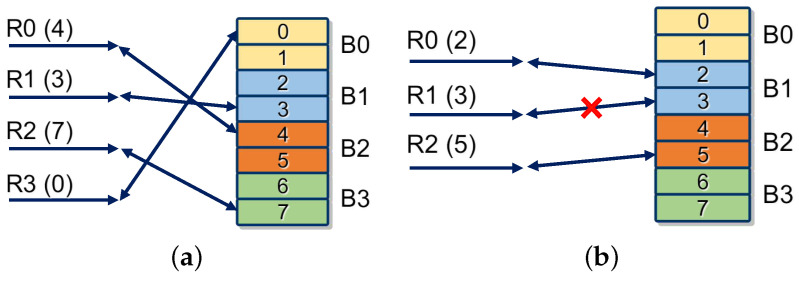
The access mode of the multiport hash table of HybriDC. (**a**) Normal parallel access, (**b**) access collision.

**Figure 8 micromachines-13-02029-f008:**
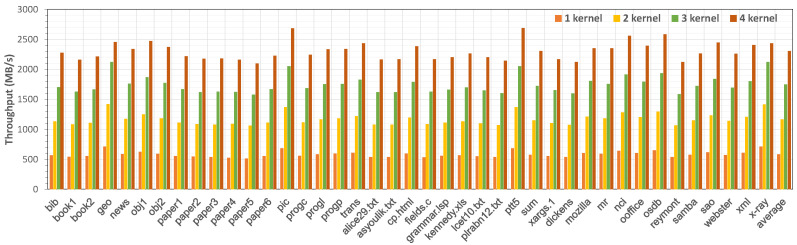
The end-to-end throughput performance of the HybriDC system with different number of compression kernels evaluated by the Calgary corpus, the Canterbury corpus, and the Silesia corpus.

**Table 1 micromachines-13-02029-t001:** Utilization proportions of various resources in different compression accelerators. The consumption proportion of the block RAM is significantly larger than those of other resources in most schemes.

Scheme	FPGA Chip ^1^	LUT	FF	Block RAM
Bartik’s [14]	XC7A100T (63,400, 126,800, 135)	1.21%	0.30%	**12.59%**
MLZ4C [15]	XC7K325T (203,800, 407,600, 445)	0.28%	0.23%	**15.51%**
Benes’ [16]	XCKU040 (242,400, 484,800, 600)	5.81%	0.58%	**13.67%**
Ledwon’s [17]	XCVU3P (394,080, 788,160, 720)	17.54%	6.31%	**36.18%**
HybriDC kernel	10AX048H2 (367,180, 734,360, 1431)	3.36%	0.56%	5.1%

^1^ The numbers of available LUTs, FFs, and block RAMs in FPGA chips are given in brackets.

**Table 2 micromachines-13-02029-t002:** Parameters related to the compression kernel design.

Symbol	Definition
(**a**) Compression algorithm parameters.	
$S_{DICT}$	Size of the dictionary
$S_{ML}$	Size of the maximum literal
$S_{HT}$	Size of the hash table
hash	Hash function
(**b**) Performance-related parameters.	
*N*	Number of integrated compression kernels
*P*	Bytes processed per cycle in one compression kernel
$T_{kernel}$	Throughput of one compression kernel
$T_{sum}$	End-to-end throughput of HybriDC
(**c**) Hardware resource parameters.	
$FF_{avl}$	Total number of available FFs
$LUT_{avl}$	Total number of available LUTs
$RAM_{avl}$	Total capacity of available block RAMs
$FF_{kernel}$	Number of FFs used in one compression kernel
$LUT_{kernel}$	Number of LUTs used in one compression kernel
$RAM_{kernel}$	Block RAMs used in one compression kernel

**Table 3 micromachines-13-02029-t003:** The compression kernel performance of HybriDC evaluated on the Calgary corpus.

Class	RAW Size (bytes)	Compressed Size (bytes)	Compression Ratio	Throughput (MB/s)
bib	111,261	55,461	2.006	539.94
book1	768,771	503,990	1.525	483.01
book2	610,856	336,337	1.816	509.02
geo	102,400	92,551	1.106	680.17
news	377,109	211,160	1.786	540.26
obj1	21,504	13,152	1.635	636.21
obj2	246,814	121,837	2.026	570.21
paper1	53,161	29,332	1.812	527.34
paper2	82,199	47,829	1.719	506.09
paper3	46,526	28,607	1.626	509.15
paper4	13,286	8533	1.557	543.17
paper5	11,954	7645	1.564	538.95
paper6	38,105	21,082	1.807	533.76
pic	513,216	88,912	5.772	704.43
progc	39,611	21,071	1.880	541.06
progl	71,646	28,599	2.505	571.98
progp	49,379	18,943	2.607	582.99
trans	93,695	31,211	3.002	614.07
Average	180,638.5	92,569.6	1.951 ^#^	562.88

^#^ 1.951 ≈ 180,638.5/92,569.6. The ratio of average raw size to average compressed size is more reasonable than the average of all compression ratios.

**Table 4 micromachines-13-02029-t004:** The resource utilization and compression performance of different LZ4 acceleration kernels.

Scheme	FPGA Chip	LUTs	FFs	Block RAM (Kbits) ^1^	Compression Ratio ^2^	Throughput (MB/s) ^3^	RAM Efficiency (Kbits/(MB/s))
Bartik’s [14]	XC7A100T	764	375	612	↓6.2%	146	4.192
Benes’ [16]	XCKU040	14,076	2803	2952	↓37.6%	760	3.884
MLZ4C [15]	XC7K325T	573	937	2484	↑2.3%	240	10.350
Xilinx [11]	XCU200	3000	3500	1908	↑1.4%	290	6.579
HybriDC	10AX048H2	12,336	4081	1460	↑1.2%	562.88	**2.594**

^1^ The block RAM used in Xilinx FPGA-based schemes was BRAM with 36 Kbits of capacity, while that used in HybriDC was Intel’s M20K with 20 Kbits of capacity. Here, we converted the number of BRAMs and M20Ks into a specific capacity to compare the on-chip memory usage. ^2^ These schemes’ compression ratios were evaluated on different datasets. For fairness, the percentage of relative difference between the hardware designs’ compression ratios and the compression ratio of standard LZ4 software was used as the evaluation criteria. Since the compression ratio of Bartik’s scheme was not disclosed, Bartik’s compression ratio was estimated based on its hash table size (4 KB) and the relationship between the compression ratio and the hash table size (Figure 4). ^3^ Bartik’s throughput was not presented in their paper. We estimated it by their design clock frequency (146 MHz).

**Table 5 micromachines-13-02029-t005:** The FPGA resource utilization comparison between the HybriDC system and the single HybriDC compression kernel.

Design	ALMs ^1^	LUTs	FFs	M20K Block RAMs	DSPs ^2^
HybriDC system (4 kernels)	34,507.8	52,955	19,157	526	64
Single kernel	7930.3	12,336	4081	73	16
Ratio	4.35	4.29	4.69	7.21	4

^1^ ALM: adaptive logic module. ALM is the basic building block in Intel FPGAs. ^2^ DSP: digital signal processor. In our design, DSPs were used to perform the hash calculation.

**Table 6 micromachines-13-02029-t006:** The power efficiency comparison between the HybriDC system and LZ4 software.

Scheme	Power (W)			Throughput	Efficiency
	CPU ^1^	FPGA	Total	(MB/s) ^2^	(MB/s per W)
HybriDC (4 kernels)	0.5×11.88	6.14	12.08	2356.65	195.09
LZ4 [18]	11.88	0	11.88	780	65.66
Ratio	-	-	1.02	3.02	2.97

^1^ The LZ4 benchmark test used one core of an i7-9700K CPU (eight cores, 95Wpower) while HybriDC used about 0.5 identical cores. ^2^ The throughput was calculated on the Silesia corpus.

## Data Availability

The data used in this study are available in [23,26,27].

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
