# Peer review of "HybriDC: A Resource-Efficient CPU-FPGA Heterogeneous Acceleration System for Lossless Data Compression"

_micromachines, 2022, doi:10.3390/mi13112029_

Round 1

Reviewer 1 Report

The authors of this paper discuss the resource imbalance problem in FPGA and present HybriDC, an adaptive resource-efficient CPU-FPGA heterogeneous acceleration system for lossless data compression. The authors discuss the issues in existing FPGA-based implementations in detail and provide performance models and measurements to compare with existing methods.

The paper contains detailed information on FPGA and lossless compression, which can be attractive to many readers of this journal. Overall I enjoyed reading this paper but found a few points that can be addressed to improve the paper's clarity further. 

First, seeing many performance numbers from different methods and hardware is great in the experiment section. However, it is hard to judge whether it is fair. I cannot pull the performance numbers (compression ratios, throughputs, etc.) directly from the referenced papers, and some numbers are said it is estimated. Can you clarify the estimated and public numbers in Table 4? Table 1 is also confusing too in that regard. How do you get such percentages? Can you make it consistent with Figure 4 to make it clear? Also, can you specify "units" in Tables 4 and 5? 

Second. In general, power analysis is a complex process. Section 5.2.4 discusses it only briefly. Can you give more details? Is it measured or estimated? How is it estimated the power consumption of CPU and FPGA? Can you describe the assumptions used? The specification of the CPU in Section 5.2.1 is different from the description in Table 6.

Lastly, it is said in 5.1.4: "In many scenarios (such

as data storage), it is unacceptable to sacrifice much compression ratio performance in exchange for throughput. In conclusion, the compression performance of HybriDC is more balanced and desirable." It doesn't sound scientific. I wonder if the authors can provide any basis (reference, etc.) or rewrite.

A few minor comments:

-. Can "2. Heterogeneous COMPRESSION ACCELERATION FRAMEWORK" be just "2. Heterogeneous compression acceleration framework"?

-. "the hardware and software parts are connected by the PCIe interface" doesn't make sense. Can it be rewritten? Something like "CPU and FPGA are connected by the PICe"?

-. There are acronyms missing definitions (e.g., ALM, DSP)

Reviewer 2 Report

Dear authors

In general terms, I consider that it is a good work and presents interesting aspects on the implementation of a lossless data compression algorithm in a heterogeneous architecture. My observations regarding the manuscript are the following:

The use of some expressions or terms in the manuscript should be revised. Some have been highlighted in the attached file.

Table I presents a comparison of works found in the state of the art with results obtained in this work. This comparison should be presented in the results or discussion section. Additionally, a comparison of the percentages of resource utilization is made without considering that devices with different characteristics are used. In this sense, the table should contain information on the amount of resources available in each device, in order to have a better perspective of the comparison.

It is important to clarify whether in related work the systems are designed to work in streaming or otherwise. In some cases, a higher on-chip memory utilization may be justified by the way data input/output is performed and the way it is processed in the kernel. This defines the parallelization strategy that can be used, with different limitations and benefits.

I am unable to identify the HDL used for the implementation of the proposed system.

It is unclear to me what is presented in Figure 4.

Is there a defined expression for CR(SD ICT, SML, SHT, hash)?

If the model is obtained experimentally, is there a table, surface plot or approximate model equation that allows to define how the optimal parameters are determined?

Latency and interval are useful measures of kernel performance and are generally presented as results in this type of work.

I am not sure that the term speedup can be used when referring to energy or power efficiency.
